



# Characteristics of surface energy balance and atmospheric circulation during hot-and-polluted episodes and their synergistic relationships with urban heat islands over the Pearl River Delta region

Ifeanyichukwu C. Nduka[1], Steve Hung Lam Yim [1,2,*], Chi-Yung Tam[3], Jianping Guo[4]

[1]Department of Geography and Resource Management, The Chinese University of Hong Kong, Sha Tin, N.T., Hong Kong, China
[2]Stanley Ho Big Data Decision Analytics Research Centre, The Chinese University of Hong Kong, Sha Tin, N.T., Hong Kong, China
[3]Earth System Science Programme, The Chinese University of Hong Kong, Sha Tin, N.T., Hong Kong, China
[4]State Key Laboratory of Severe Weather, Chinese Academy of Meteorological Sciences, Beijing 100081, China

Corresponding author: Steve H.L. Yim (steveyim@cuhk.edu.hk)

**Abstract.** This study analyzed the nature, mechanisms and drivers for Hot-and-Polluted episodes (HPEs) in the Pearl River Delta, China. Numerical model simulations were conducted for summer and autumn of 2009–2011. A total of eight HPEs were identified, mainly happening in August and September. K-means clustering was applied to group the HPEs into three clusters based on their characteristics and mechanism. We found three HPEs were driven by weak subsidence and convection induced by approaching tropical cyclones (TC-HPE), two HPEs was controlled by calm conditions (ST-HPE) with low wind speed at the lower atmosphere, whereas the remaining three HPEs were driven by the combination of both aforementioned systems (HY-HPE). TC-HPE and ST-HPE had both positive synergistic effect between HPE and UHI (~1.1°C increase); whereas no discernible synergistic effect was found in HY-HPE. Total aerosol radiative forcing (TARF) caused a reduction in temperature (0.5–1.0°C) in TC-HPE and ST-HPE, but an increase (0.5°C) in HY-HPE.

## 1 Introduction

Air pollution and heat waves were identified as major atmospheric environmental disasters (Khafaie et al., 2016; Xia et al., 2018). Air pollution can vary at spatial scales ranging from local to global (Shi et al., 2020; Wang, et al., 2019; Wang, et al., 2019; Yang et al., 2019; Yim et al., 2009; Yim et al., 2014; Yim et al., 2010, 2015). Studies have reported the significant effects of air pollution on human health (Chen et al., 2017; Gu et al., 2018; Liu et al., 2017), ecosystems and biodiversity (Lovett et al., 2009; Nowak et al., 2015; Yim et al., 2019b), and weather and climate (Guo et al., 2016, 2018; Z. Liu et al., 2018; Z. Liu et al., 2019, 2020).

Heat waves are generally defined as extended periods of elevated temperatures across the globe varying in frequency, intensity, and duration. Previous studies have revealed the significant impacts of heat waves on human health and mortality





(Matusick et al., 2018), and natural systems (Unal et al., 2013). More seriously, heat wave events are likely to occur more

frequently in future years as a consequence of climate change (Murari & Ghosh, 2019; Silva et al., 2016; Wang et al., 2018). Despite their large spatial heterogeneity, air pollution and heat waves have been shown to share several common underlying meteorological drivers (Founda & Santamouris, 2017; Papanastasiou et al., 2014; Zhang et al., 2017). Although great advances have been made in either air pollution or heat wave, few studies have comprehensively investigated both hot-and-polluted episodes (HPEs) in which extreme temperature and air pollution occurred simultaneously, causing an even more

serious impact on human health due to their synergic effect (Pinheiro et al., 2014; Qian et al., 2010; Scortichini et al., 2018). Before this effect can be comprehensively assessed, fully understanding the driving mechanisms underlying the formation and remediation of HPEs is imperative.

The processes responsible for HPE formation and intensity vary case by case (S. Fan et al., 2011; Katsouyanni et al., 1993; Ordóñez et al., 2010; Yim, 2020). For instance, Yim, (2020) analyzed the atmospheric condition in 3D during this events,

and found that HPEs were mostly associated with a reduction in both vertical and horizontal wind velocities. By contrast, (S.. Fan et al., 2011) illustrated that approaching tropical cyclones, which may cause subsidence, weak vertical diffusion, and poor horizontal transport, were the mechanism responsible for the HPEs. To obtain a complete picture of all possible mechanisms of HPEs, a multi-episodic study is therefore warranted.

The surface energy balance, which modified and demarcated turbulent energy fluxes, were well illustrated to influence heat

waves (Founda and Santamouris, 2017; Li et al., 2015; Li and Bou-Zeid, 2013; Miralles et al., 2014). For instance, Miralles et al. (2014) found that soil desiccation led to the modification of turbulent heat fluxes through the changes of latent and sensible heat fluxes. Li and Bou-Zeid (2013) and Li et al. (2015) demonstrated the effect of built-up and vegetated surfaces on the demarcation and modification of turbulent energy. However, the interaction between surface characteristics and atmospheric conditions during HPEs, which may exhibit synergies, has yet to be completely understood. The role of surface

characteristics in HPE formation and development merits further investigation.

Studies are limited in China and more far in-between in the PRD region. Previous HPE related studies in China has been focused mostly on summertime $O_3$ mechanisms and characteristics (Gong and Liao, 2019; Lam et al., 2005; Li et al., 2018b; Shu et al., 2020), while others focused on atmospheric drivers for, and interactions between air pollution and mortality. However, other studies analyzed the atmospheric boundary layer characteristics over PRD using measurements and

numerical models to identify boundary layer conditions that could result in HPE (S. Fan et al., 2008; S. Fan et al., 2011; Yim, 2020).Thus far, there has been limited research on HPE mechanisms in China, particularly in the Pearl River Delta (PRD) region, which has been rapidly and substantially urbanized in recent years (Li et al., 2016). The urbanization has a significant impact on the regional climate and air quality through modification of the ecosystem, land surface, atmospheric, and energy processes (Mirzaei and Haghighat, 2010; Wang et al., 2019, 2020; Xie et al., 2016; Yim et al., 2019c; Yu et al.,

2014). Mirzaei and Haghighat (2010) identified the ways urbanization modified the surface cover, climate, and energy processes, including the conversion of more surfaces into urban impervious surfaces; alteration of the local winds; humidity, temperature, and precipitation patterns; and changes in demarcation in turbulent energy fluxes within the surface and




boundary layer. Furthermore, PRD is susceptible to events associated with the monsoon and tropical cyclone activities that may cause HPEs (Fan et al., 2011; Li et al., 2018), making the PRD region an ideal location for assessing HPEs.

This study identified all the HPEs occurring during 2009–2011, and analyzed their associated thermodynamic and circulation characteristics. Representative episodes were examined for a possible synergistic relationship with urban and vegetated land covers within the PRD region (see SI section 1 and Figure S2 for details in land cover characteristics, and delineation of urban and vegetated surfaces). This is expected to end up advancing knowledge regarding the factors responsible for the evolution and sustenance of HPEs as well as the relationship between HPEs and surface characteristics.

## 2 Materials and Methods

### 2.1 Observations

This study used meteorological and air quality measurements for model validation and HPE identification. Hourly air temperature at 2 m above ground ($T_2$) for the study was obtained from Hong Kong Observatory (HKO). Hourly mean concentration of coarse particulate matter ($PM_{10}$) and ozone ($O_3$) concentrations was obtained from Environmental

Protection Department (EPD - https://cd.epic.epd.gov.hk/EPICDI/air/station/?lang=en). The hourly data were used for model validation, whereas daily maximum and mean values were calculated based on the hourly data for HPE identification.

### 2.2 HPE Selection

The concept of hot polluted episodes that refer to an episode with coincident high temperature and air pollution level have been investigated (Katsouyanni et al., 1993; Ordóñez et al., 2010; Yim, 2020a), however, most of the studies have been

focused on their compounding impact on health or the impact of the excess temperature on air pollution levels. Only few studies (Fan et al., 2011; Yim, 2020a) have considered the mechanisms responsible for their co-occurrence. While the previous study focused on high temperature and $PM_{2.5}$ pollution in Hong Kong, this study focused on high temperature and high $PM_{10}$ and $O_3$ in the PRD region. Similar to Yim (2020), this study identified a HPE based on daily maximum temperature, and daily mean of $PM_{10}$ and $O_3$. $PM_{10}$ was used in this study because of the lack of $PM_{2.5}$ data in the study

period which was June to October in 2009–2011. The time period was selected because these months represent a period in a year with the highest daily maximum temperature, which when combined with days with poor air quality, forms a HPE. $T_2$, $O_3$ and $PM_{10}$ thresholds were first defined as their 50th percentiles at each station in the study period. The justifications for the threshold is presented in section 3 of SI. A HPE was identified when, daily maximum temperature on each day exceeded its threshold for at least three consecutive days, and daily mean $O_3$ and $PM_{10}$ levels on the same days exceeded their

threshold values at more than 50% of all air quality stations. All identified HPEs corresponded to a significant regional climatic event that past studies (Freychet et al., 2017; Hou et al., 2019; Yim et al., 2019a) have suggested could lead to either an increase in temperature, a deterioration in air quality or both. It is noted that the thresholds derived here and those derived



in Yim (2020) were different. It is because this model study was designed to focus on the most extreme cases to balance between the science and computational cost.

## 2.3 Model Description and Setup

This study employed Weather Research and Forecasting (Skamarock and Klemp, 2008) with Chemistry (WRF-Chem) version 3.7.1 (Grell et al., 2005), which is a nonhydrostatic, mesoscale numerical model coupling both meteorology and chemistry. This model downscaled the meteorology and air quality in three domains at a downscaling ratio of 3 (Figure S1a in the SI): domain 1 (D1; 27 km) covering the whole of China, domain 2 (D2; 9 km) covering southern China, and domain 3 (D3; 3 km) covering the PRD region. The detailed model configuration is provided in Section 1 in SI. The initial condition data used was provided by NCEP Final Analysis (FNL from GFS) with a $1° \times 1°$ resolution, while the boundary condition was updated every 6 hours from FNL data. The emissions data for Hong Kong were provided by the Hong Kong Environmental Protection Department (HKEPD), whereas the emissions within PRD except HK were provided by Zheng et al (2009). The emissions outside the PRD region were based on the INTEX-B 2006 regional emission inventory (Zhang et al., 2009). Biogenic emissions were based on Guenther et al (2006), whereas shipping emissions were based on Streets et al (2003).

We performed a series of WRF-Chem simulations for each HPE. The performance of the model was evaluated against observations, which was detailed in Section 2 of SI. For each HPE, two sets of WRF-Chem simulations were conducted with a 2-day spin-up period. Two additional days were included before and after the HPEs in each simulation run. This setting was used to identify and calculate trends in the variables before, during, and after HPEs. The first set of simulations turned on the aerosol–radiation feedback option while the second turned off the option. The difference between the two sets of simulations was attributable to the effect of aerosols on radiation fluxes during an HPE. The results were separated into groups using K-means clustering, and a representative of each group characterized based on the mechanisms responsible for its formation. The mechanisms of, thermodynamic and circulation patterns of HPEs, and synergistic relationship between HPEs and surface cover were then discussed in detail.

### 2.4 HPE Clustering

Based on our identification method, eight HPEs were identified (Table S4). To enhance our understanding of atmospheric conditions in each HPE, cluster analysis were performed of meteorological variables (Stefanon et al., 2012; Tan et al., 2019) such as $T_2$, Sea Level Pressure (SLP), specific humidity at 2 m, wind components (u, v) at 10 m, incoming solar radiation at the surface, and geopotential height (GPH) at 500 hPa, which could group these variables from different times into clusters having the same meteorological conditions. A modified K-means clustering algorithm (Hartigan and Wong, 1979) function on NCAR command language (NCL) with its center set at random and iterations set at 1,000,000 was applied for all the aforementioned meteorological variables.



## 3 Results and Discussion

### 3.1 HPE Identification and Classification

Due to the different durations of HPEs, mean diurnal variations of the variables for the study domain during the three periods (before, during, and after) from the model results were analyzed. The dissimilarity was optimized at three clusters showing the number of groups that the HPEs could be grouped into. The results of the cluster analysis indicate that the HPEs were classified based on their formation and characteristics into: Tropical Cyclone (TC-HPE); Stagnant (ST-HPE); and Hybrid

(HY-HPE), and they are discussed below. The weather conditions of the TC-HPE were characterized by lower SLP and GPH but higher $T_2$ and specific humidity. The TC-HPE typically occurred when a tropical cyclone was approaching the PRD region. A representative episode (TC-HPE$_{rep}$) occurring from the August 29 to September 1, 2010, was selected to further explain the weather conditions along with weather charts provided by the Hong Kong Observatory (HKO; see Figure S4 in SI). In this episode, the tropical storm LIONROCK passed through the PRD region. Its passage caused stagnation and

expansion of the high-pressure system in the lower atmosphere over the north of the region. This led to an increase in $T_2$ over the region. Mid-day $T_2$ in Hong Kong rose from 32 °C on August 29 to 34 °C on August 31, which broadly corroborated our previous studies (Yim et al., 2019b) that tropical cyclones within 1,100 km of the region can cause HPEs.

The second group, ST-HPE, was found to be characterized by higher temperature and lower specific humidity in the PRD region, as demonstrated by the representative HPE (ST-HPE$_{rep}$) occurring from August 4 to 7, 2010 (see Figure S5 in SI).

This group represents a slow-moving weather system formed by a low-pressure system covering most of China and the Philippine Sea. This is in contrast to previous studies that have reported a strong association between high-pressure systems and air stagnation. However, when Freychet et al. (2017) assessed the dynamical processes for heatwave formation over Eastern China, they found that a low-pressure system along with northerly flow within the region often led to heat convergence and an extreme high temperature episode. The ST-HPE$_{rep}$ was observed to have a region-wide slow-moving

weather condition characterized by a wide low-pressure system that covered most of East Asia for a few days.

The third group, HY-HPE, has a unique profile that was defined by Karremann et al. (2016). This group consisted of more than one dominant synoptic condition during an episode. A representative HY-HPE (HY-HPE$_{rep}$) occurring from September 7 to 10, 2011, was compared with corresponding HKO charts (see Figure S6 in SI). The result confirmed the co-occurrence of more than one large scale weather system during the HPE. The result of their characterization demonstrated that different

pressure centers, synoptic conditions, or their combination can be juxtaposed during a weather event and that their gradient will determine the weather condition of a region.

### 3.2 HPE Characterization

This section mainly analyzes the formation mechanism for each HPE group. Due to the similarities of the members in each group, we discuss the air quality, energy, and circulation characteristics of one representative HPE from each group.




Figure 1. Air temperature at 2m ($T_2$; red), cloud fraction (CF; yellow), and incoming shortwave radiation (blue) at ground level for (a) TC-HPE, (b) ST-HPE, and (c) HY-HPE; $PM_{10}$ (red), $O_3$ (yellow), outgoing longwave radiation (OLR) at the top of the atmosphere (blue) for (d) TC-HPE$_{rep}$, (e) ST-HPE$_{rep}$, and (f) HY-HPE$_{rep}$; Time series of vertical velocity (color shading) for (g) TC-HPE$_{rep}$, (h) ST-HPE$_{rep}$, and (i) HY-HPE$_{rep}$. Positive vertical velocity means downward motion, whereas

negative vertical velocity means upward motion. Black boxes represent the period of HPEs. Days 1 and 2 were before the HPEs; days 3–6 were during the HPEs; and days 7 and 8 were after the HPEs.



### 3.2.1 TC-HPE

Figure 1a depicts the time series for domain averaged hourly $T_2$ during the $HPE_{TC}$ representative HPE. During the TC-$HPE_{rep}$, $T_2$ increased on day 3 [onset (~30.2 °C)], reached a peak (~32.7 °C) on day 5, and decreased on day 7 (~30.8 °C).

Figure 1g shows the height-time cross-section of vertical velocity. Positive vertical velocity means downward motion, whereas negative vertical velocity means upward motion. In the lower atmosphere, the positive and even relatively small negative vertical velocity values during the TC-$HPE_{rep}$ indicate a weak subsidence of air masses resulting from an approaching tropical cyclone (Luo et al., 2018; Yim et al., 2019a). As shown in Figure 1a, the cloud fraction was maintained at approximately 0.40 at the onset of the TC-$HPE_{rep}$ and reduced to 0.20 at 08:00 on day 5. The reduction in cloud fraction

was due to weak convection. These clear sky conditions caused a significant increase in incoming shortwave radiation, leading to a remarkable increase in $T_2$ due to greater direct exposure to solar radiation at the surface. Figure 1d shows that the outgoing longwave radiation (OLR) also increased up to 250 W/m² at 08:00 on day 5. The situation changed by the end of day 6 due to the increased upward motion, as indicated by the large negative vertical velocity shown in Figure 1g. A greater number of convective activities led to an increase in cloud fraction (up to 0.30) and thus a reduction in SW radiation.

Consequently, $T_2$ decreased, signaling the end of the HPE.

During TC-$HPE_{rep}$, the weak vertical motions led to an accumulation of $PM_{10}$, as shown in Figure 1d. The $PM_{10}$ concentration rose from approximately 10 μg/m³ on day 3 to >24 μg/m³ on day 5. This change can be attributed to the northerly wind during the episode, as shown in Figure S7a in SI. Enhanced northerly wind has been found to significantly enhance transboundary air pollution in the region (Hou et al., 2019; Luo et al., 2018; Tong et al., 2018b; Yim et al., 2019a).

During the TC-$HPE_{rep}$, $O_3$ accumulation also occurred. The increase in insolation due to lower cloud fraction and accumulation of pollutants, some of which are precursors of $O_3$ formation, led to the increase in mid-day $O_3$ concentration from approximately 60 μg/m³ on day 3 to approximately 90 μg/m³ on day 6. The result also demonstrated that the diurnal variation in $O_3$ concentration was maintained but the mean concentration increased significantly as the TC-$HPE_{rep}$ got hotter and more polluted before falling at the end of the episode.

### 190 3.2.2 ST-HPE

Figure 1b depicts the evolution of the ST-$HPE_{rep}$. $T_2$ increased on day 3 [onset (~32.7 °C)], reached a peak (~33.3 °C) on day 4 and then decreased until the end of day 6 (~32.1 °C). The ST-$HPE_{rep}$ was distinguished by a 5 m/s wind speed at the lower atmosphere from day 1 to 4, as shown in Figure S7b in SI. The wind speed reduced to approximately 2 m/s on day 3, which led to an intense accumulation of heat and a 1.7 °C increase in $T_2$. The subsequent increase in wind speed after day 4 (>5

m/s) coincided with the decrease in $T_2$ until the end of day 6, which marked the end of this episode. Vertical movement during this event, as shown in Figure 1h, ranged from −0.1 to 0.2 m/s. The strong subsidence observed on days 2 and 3 in the mid/lower atmosphere contributed to the stable atmosphere because the cloud fraction (shown in Figure 1b) was lower (~0.20 on day 2, whereas ≤0.1 on day 3). On day 4, the convective motions (−0.1 m/s) contributed to a steady increase in





cloud fraction to 0.5. Consequently, the increase in cloud cover cut off insolation (down to approximately 1,700 W/m$^2$ on

day 4 and approximately 1,550 W/m$^2$ on day 5, as shown in Figure 1b). However, the cloud could retain most of the heat on

day 4 as the OLR displayed in Figure 1f decreased from 210 W/m$^2$ on day 3 to 150 W/m$^2$ on day 4, while the T$_2$ increased by

0.6 °C. T$_2$ reduced thereafter by 2.3 °C due to the amount of persistent high cloud, which blocked incoming shortwave

radiation and cut off the source of heat. This episode persisted until the end of the ST-HPE$_{rep}$ (day 6).

The calm wind condition also contributed to the deterioration in air quality, as shown in Figure 1e. The increase in the

concentration of PM$_{10}$ to 20 µg/m$^3$ from days 4 to 6 was accentuated by the change in wind direction to north/northeast, as

shown in Figure S7e in SI. This result was consistent with a number of findings (Hou et al., 2019; Luo et al., 2018b; Tong et

al., 2018b, 2018a; Yim et al., 2019b) showing that a northerly wind is usually associated with air pollution in the region as a

result of transboundary air pollution. The increase in O$_3$ concentration displayed in Figure 1e is attributable to the increase in

precursors and the amount of incoming shortwave radiation available. The diurnal variation in the time series of O$_3$ is also

indicative of this cumulative trend.

### 3.2.3 HY-HPE

Compared the other two groups, HY-HPE did not show an obvious increase in daily T$_2$ during the episode. As shown in

Figure 1c, the HY-HPE$_{rep}$ (days 3–6) had a mean daily maximum T$_2$ of approximately 30.0 °C with a variation of only ±0.3

°C. The HY-HPE$_{rep}$ showed a relatively large change in horizontal wind speed at the lower troposphere (Figure S7c in SI)

from <5m/s on days 1 to 3 to a higher wind speed (≥8 m/s) from days 4 to 8. Similar to the other two groups, Figure 1i

shows a weak subsidence in this episode due to weak vertical velocity (from −0.2 m/s to 0.2 m/s), which was accompanied

by weak downward motion (except on day 5). The increased horizontal wind speed from day 4 contributed to the

temperature regime of the HY-HPE$_{rep}$ by offsetting the heat buildup due to the weak vertical motion. This led to marginal

variations in the diurnal temperature between pre-, mid-, and post-HPE days.

The changes in air quality during this episode were influenced primarily by the horizontal wind direction and speed. This is

exemplified by the effect of the prevailing wind speed and direction on the relatively lower concentration of PM$_{10}$ (about 15

µg/m$^3$ at its peak; see Figure S7i&f in SI). The prevailing easterly wind suggests a marginal effect of transboundary air

pollution in HY-HPE; the increase in air pollutant level was thus due to local emissions. The subsequent increase in wind

speed resulted in a dispersion and reduction in the PM$_{10}$ concentration. However, as the wind direction switched to the north

(day 6), PM$_{10}$ concentration began to rise due to increases in transboundary air pollution. Changes in O$_3$ concentration

followed the same pattern due to the availability of precursors and the influence of wind speed and direction. Consequently,

HY-HPE was unable to accumulate heat, PM$_{10}$, and O$_3$ concentration despite the weak convection.

### 3.3 Synergistic Relationship

We examined the synergistic relationship between UHIs and HPEs during these episodes. This was achieved by scrutinizing

the daily mid-day height-time series of potential temperatures, which have been found to be a suitable indicator for heatwave





accumulation and structure because they exhibit the characteristics of an air parcel over time (Miralles et al., 2014; Ramamurthy et al., 2017). OLR was also adopted as an indicator to show how transparent the sky was at the time. The effects of land cover on the evolution and sustenance of the HPE were analyzed by examining UHI intensities during the HPEs. The effect of the air quality on the HPEs was also examined, largely through analyzing the effect of total aerosol

radiative forcing (TARF).





### 3.3.1 TC-HPE

Figure 2. (a-c) Difference of $T_{2(diff)}$, sensible heat ($SH_{(diff)}$), and latent heat ($LH_{(diff)}$) between urban and vegetated areas for TC-HPE$_{rep}$, ST-HPE$_{rep}$, and HY-HPE$_{rep}$; (d-f) Vertical profile of potential temperature ($\theta$) for TC-HPE$_{rep}$, ST-HPE$_{rep}$, and HY-HPE$_{rep}$ at mid-day for the PRD region; (g-i) the changes of $\Delta T_2$, $\Delta CF$, and incoming shortwave radiation ($\Delta SW$) at the surface induced by total aerosol radiative forcing for TC-HPE$_{rep}$, ST-HPE$_{rep}$, and HY-HPE$_{rep}$.



Figure 2a shows the $T_2$ difference between urban and vegetated areas during TC-HPE$_{rep}$. The results indicate a remarkable 3.9 °C to 5.5 °C difference in $T_2$ between the two types of land cover due to the UHI effect. Figure 2d depicts the difference
between the potential temperature over urban and vegetated land covers in the PRD region at mid-day. The temporal distribution in potential temperature shows an increasing trend in the lower atmosphere (≥1.8 °C from days 4 to 6) and upward expansion in height for warm air beyond 1.2 km (0.3 °C on day 5). Persistent daily heating from incoming solar radiation provided energy that precipitated a rise in temperature, thereby entrapping and accumulating heat during the TC-HPE$_{rep}$. For the TC group of HPEs, the $T_2$ difference ranged from 0.5 °C to >1.5 °C, showing that TC-HPE provided extra
heat to the surface which enabled the urban areas to warm up to a greater extent than normal. This result indicates that the UHI effect was enhanced during the TC-HPE.

The effect of surface moisture availability, surface energy retention, and transfer for both urban and vegetated surfaces was investigated by analyzing latent and sensible heat fluxes (Figure 2a). The largest absolute latent heat difference between the two types of land cover was between 350 and 400 W/m$^2$. This illustrates the impact of the limited vegetation in the city. This
is buttressed by the 130–200 W/m$^2$ difference, respectively in the amount of sensible heat for both types of land cover. While both the latent and sensible heat fluxes shared similar incoming shortwave radiation, the fact that the sensible heat results were lower than the latent heat results suggest that more water was available for evapotranspiration due to the tropical cyclone-associated rainfall.

TARF caused an initial increase in $T_2$ but then a significant reduction in $T_2$ (>1.0°C) on day 6 when the PM$_{10}$ increased to a
peak. Figure 1d and Figure 2g show that when PM$_{10}$ was increased but below 10 μg/m$^3$, the TARF caused a reduction in cloud fraction and thus an increase in $T_2$. The reduction in cloud fraction due to TARF may be due to the aerosol-cloud interaction in which aerosols served as cloud condensation nuclei and thus more cloud droplets were formed with a smaller radius, and the cloud fraction hence decreased (Liu et al., 2020). Nevertheless, when the PM$_{10}$ level exceeded 10 μg/m$^3$, the aerosol-radiation interaction (aerosol scattering and absorption) may become dominant, causing a reduction in amount of
solar radiation reaching to the ground level due to the blockage effect of aerosols and thus temperature reduced.

### 3.3.2 ST-HPE

Figure 2b shows the difference in $T_2$ between the two land covers in the ST-HPE$_{rep}$. The $T_2$ in urban areas was higher by a range of approximately 2.8 °C–4.0 °C during the ST-HPE$_{rep}$. There was also approximately 1.0 °C difference in $T_2$ between the ST-HPE$_{rep}$ period and its pre/post periods. Figure 2e shows that heat accumulated before the onset of ST-HPE$_{rep}$ began to
decay during the episode. The heat accumulation led to a maximum difference of 1.5 °C on day 3, which was the first day of the ST-HPE$_{rep}$ and the day with the highest incoming solar radiation. Comparing urban and vegetated land covers, there was a difference of approximately 250 and 400 W/m$^2$ in the amount of latent heat and sensible heat, respectively (Figure 2b). The potential temperature also shows a positive synergy during the ST-HPE$_{rep}$ by maintaining a positive difference throughout the episode, even though it progressively weakened over time. These results indicate that the UHI effect was accentuated
during the ST-HPE$_{rep}$, leading to a larger temperature difference between urban and vegetated surfaces.





The results of the effect of TARF on $T_2$, $SW_{in}$, and CF for ST-HPE indicate that TARF caused marginal changes during the pre- and post-ST-HPE periods (Figure 2h). Like the TC-HPE$_{rep}$, the TARF effect led to a reduction in approximately 1.5°C when the $PM_{10}$ reached a peak on days 4 and 5. The cloud fraction (Figure 1b) also increased significantly during this period, thus contributing to the decrease in $T_2$. As explained in the TC-HPE$_{rep}$, the reduction in $T_2$ could be mainly due to effect of
aerosol–radiation interaction. The TARF effect on $T_2$ by day 6 led to a 0.5 °C increase: this was due to the slight increase of $PM_{10}$ concentration (the total level was lower than 8 µg/m$^3$) on that day that reduced the cloud fraction.

### 3.3.3 HY-HPE

HP-HPE$_{rep}$ did not show any significant difference in UHI intensity between the pre-, mid-, and post-HPE periods. Figure 2c shows a similar magnitude (3.0–3.5 °C) during, pre- and post-HPE periods. The daily variation did not display any obvious
signs of heat retention in the atmosphere, indicating that there was no heat accumulation during this episode. The potential temperature shown in Figure 1c indicates that heat generated from heating the surface by the incoming solar radiation could not be entrained due to its increased wind speed and weak vertical motions, thereby making it difficult to accumulate heat; an observed pre-requisite for HPE and UHI synergistic effect. There was therefore no synergistic relationship between UHI and HPE during the HY-HPE.
The effect of TARF on $T_2$ and OLR in the HY-HPE$_{rep}$, as displayed in Figure 2i shows a variation of −0.5 °C to 0.5 °C in temperature across the pre-, mid-, and post-HPE periods. The pre-HPE period shows marginal changes for CF, $SW_{in}$, and $T_2$; however, variations began to be apparent from day 5. For instance, on the afternoon of day 5, the $PM_{10}$ started to increase and reached a peak early on day 6. The cloud fraction increased in response to the increase in $PM_{10}$, as did $SW_{in}$ and then $T_2$. These changes became marginal late on day 6. Nevertheless, $PM_{10}$ still maintained a relatively high level and the cloud
fraction reduced once more, allowing more solar radiation to reach the ground causing an increase in $T_2$.

### 4 Conclusions

In this study, eight HPEs during 2009–2011 were identified. We found that TC-HPE was influenced by incoming solar radiation and subsidence, ST-HPE influenced by low wind speed and reduction in cloud cover; while HY-HPE was influenced by weak vertical transport, and increased wind speed. Consequently, TC-HPE and ST-HPE had a positive
synergetic relationship (~1.1 °C) between UHI and HPE because their characteristic meteorological conditions (weak subsidence, low wind speeds and reduced cloud cover) accentuated insolation, heat storage and heat entrainment, while HY-HPE had no discernable relationship. TARF was found to contribute significantly to the temperature variations during the HPEs, leading to significant cooling effects on the HPEs (0.5–1 °C), except when upward vertical transport prevailed, particularly during the daytime.
This study provided further insight into the nature of HPEs at both regional and local level by analyzing their modes of formation and thermodynamic characteristics, and by investigating the contributions of land use/land cover (LULC) changes



to the nature of HPEs. These findings can aid policymakers in redesigning and renewing our urban environments to alleviate the UHI effect, global warming, and air pollution. Especially the impact of aerosol radiative forcing during the HPEs will improve our understanding of the mechanisms responsible for the co-occurrence of unusually high temperature and poor air

quality in the PRD region, showing enormous implications for regional climate and health. The next phase of our study will focus on the regional scale implications of the effect of HPEs on human health, the relative contribution of aerosol–radiation and aerosol–cloud interactions, and the combined effect of LULC and aerosols on our climate.

**Data availability:** The in-situ meteorology data was provided by Hong Kong Observatory at

http://www.hko.gov.hk/cis/climat_e.htm (last access: 14 September 2019); while the in-situ air quality data was provided by Hong Kong Environmental Protection Department at https://cd.epic.epd.gov.hk/EPICDI/air/station/?lang=en (last access: 14 September 2019).

**Supplement:** The supplement related to this article is attached.

**Author contributions:** Steve Yim and Ifeanyichukwu C. Nduka designed the study. Ifeanyichukwu C. Nduka did all

calculations with support from Steve Yim, Chin-Yung Tam. Ifeanyichukwu C. Nduka wrote the paper with support and editing from Steve Yim, Jianping Guo and Chi-Yung Tam.

**Competing interests:** The authors declare that they have no conflict of interest.

**Disclaimer:** The views expressed in this paper are those of the authors and do not necessarily reflect the views or policies of the Hong Kong government or any of its agencies or parastatal.

**Acknowledgments:** This work is supported by the Early Career Scheme of Research Grants Council of Hong Kong (grant no. CUHK24301415). The in-situ meteorology and air quality observations are provided by Hong Kong Observatory (http://www.hko.gov.hk/cis/climat_e.htm) and Hong Kong Environmental Protection Department (https://cd.epic.epd.gov.hk/EPICDI/air/station/?lang=en)

**Financial support:** This work is supported by the Early Career Scheme of Research Grants Council of Hong Kong (grant

no. CUHK24301415).

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
