# Peer review of "Characteristics of surface energy balance and atmospheric circulation during hot-and-polluted episodes and their synergistic relationships with urban heat islands over the Pearl River Delta region"

_Atmospheric Chemistry and Physics, 2020_

## Referee Comment (RC1) · Anonymous Referee #3 · 2 Jan 2021

General comments The study focuses on the region of Pearl River Region (China) and investigates hot and polluted episodes (HPEs) from 2009 to 2011, classified in relevance to atmospheric circulation, while it explores variability of surface energy balance parameters in an effort to detect possible synergies between HPEs and urban heat island (UHI). Numerical simulations using WRF-Chem were conducted in the domain, after evaluating model performance through observations. The subject of the study is relevant to the thematic areas of the journal. Synergistic effects between heat waves,

urban heat islands and air pollution are of particular importance in the general context of climate change and environmental threats. Although the subject of the study is interesting, the aim, specific objectives and novelty of this study should be further highlighted. Interpretation and discussion of the results remains somewhat superficial, especially regarding the role of different surface fluxes during HPE (compared to pre- and post – HPEs episodes) and how they contribute to positive feedbacks between HPEs and UHI strength in the study area. Methodological approaches need further clarification and precision in several parts of the study (in which authors seem to speculate) as for instance adopted criteria for the identification of HPEs and utilization of meteorological and air quality observations from the available stations network in the study area. Limitations of this study should be included in the discussion. Specific comments are provided below, which I hope will be helpful to the authors Specific Comments Why the period of 2009-2011 was used? Recent decade has witnessed extreme hot weather and could be more representative of recent and near future tendencies. Also, I missed some qualitative and quantitative information about the observed air quality and temperature trends in the area during the last years. For instance, are heat waves, polluted episodes, or combination (HPEs) becoming more frequent? Line 49-50 : actually the outcome of these studies is that asymmetric changes in the surface energy budget equation terms during heat waves (compared to normal summer conditions) , have been found to induce synergies between heat waves and UHI magnitude Line 70 – 75: The title of the manuscript refers to synergies between HPEs and UHIs. But here, the main aim / objectives and intentions of the study are not clearly stated. Line 86-87: what previous study? Please specify and rephrase. If you want to stress the innovation of the 'current ' study which does not duplicate a 'previous' one, just underline what has been already done and what is new in this study. There is a confusion with citations and reference list of Yim. Yim 2020, Yim 2020a and Yim 2020b, all 3 appear in the text. In the reference list they appear as 2020a and 2020b, but these two references are also identical. Please clarify and correct.

The criteria for the definition and methods for identification of HPE should be better presented and/or justified. There is a plethora of heat waves (or hot episodes) definitions in literature based either on percentiles of maximum temperature distribution (reflecting local climate conditions), fixed temperature thresholds or combination. A common duration of 3 days is usually adopted. According to the authors, the 50% percentile of T2 exceeds the critical temperature above which risks of health impact were significant (SI, line 106). Is it for Hong Kong station? Which is this threshold? This means that at a frequency of 50% HK experiences temperatures above critical values? The same for air quality thresholds. Provide some quantitative information. Are WHO thresholds exceeded at a frequency of 50% in the area? Is it for HK stations (being also cleaner than other air quality stations, according to authors?). Îd'he authors should proceed for their identification over the study period based on a more robust definition of HPEs. In the sensitivity test (Table S3 in SI) I had the impression that the percentile 50% was selected just to secure a 'good' number of HPEs (not too few, not too many?). What is the measure of the 'optimum' number of HPEs? In general the authors must be and look more confident about their choices and procedures. Lines 92-95. Here it is reported that for HPEs identification, maximum air temperature exceeded threshold at each station ( I suppose the authors refer to 16 meteorological stations within PRD region, as shown in map S1b) and PM10, O3 exceeded thresholds for 50% of all air pollution stations (I suppose from the 11 air quality in total as also reported in supplement in PRD region) . But in Supplement, (section 2.2), HPEs identification seems to be based only on one station's (Hong Kong) temperature, and on 14 air quality stations within Hong Kong. Perhaps I misunderstood something, but I think it is somewhat confusing. In each case, the approach needs further clarification. Line 97-99: I do not understand the point, it is very general statement. Justify better your selection criteria. Are they more strict or less strict compared to previous studies and why? Computational cost is important, but scientific base is more important. Section 2.1 – observations. The number of stations should be mentioned here along with reference to supplement for more details on locations etc. Line 104: write 'in Supplement Information' instead of SI when first mentioned in the text. Line 106: provide links Line 115: 'trends' ? Line

118: rephrase Line 120: .. 'and synergistic relationship between HPEs and surface cover': please rephrase the part of 'surface cover' Line 199: As you refer to short wave radiation (SW), the values 1,700 and 1,500 W/m2 are unrealistic here, based also on the scale of SW. Section 3.2.3 (HY-HPE): according to criteria set for T2 , the identification of HPE requires T2 > 50% percentile, corresponding to 31.3 0C (as shown in table S3), but simulated temperatures are lower, not corresponding to heat wave criteria. In Line 218, the authors also report marginal changes in pre- mid- and post HPE, which means that no heat wave is identified in this case. Please comment on this. Provide some information/comparison between observations and simulations for the particular HPEs which you have selected to demonstrate. In each case, Figure captions should be more informative and indicate whether the results refer to WRF simulations or observations. Line 230: '. . .which have been found to be a suitable indicator for heatwave accumulation and structure because they exhibit the characteristics of an air parcel over time' ??? ' I do not understand the statement that potential temperature exhibit the characteristics of an air parcel over time . Please rephrase/correct! Line 233: OLR is mentioned here (also in line 290 ) but I cannot see how it is used in the analysis and results described in this section. Line 234: How were UHI intensities quantified? Do they correspond to T2 differences between 'average urban' grids and 'average rural' grids? (the same for sensible and latent heat differences) . How do they link with map of Figure S2? Please provide relevant information. What about observed UHI intensities between urban and rural stations during HPEs? Provide an example. Are there synergies? Line 238 (figure caption): replace 'Difference of T2(diff)' with ' Difference of T2 (T2(diff))' In the analysis and investigation of possible 'synergies' between UHIs and HPEs, you should better and further emphasize the 'differences' of UHI magnitude between HPE and no-HPE conditions (before and after the episodes). If and how UHI intensity is exacerbated (or possibly attenuated) during the episodes and how it links with possible corresponding differences in shortwave radiation, latent or sensible heat fluxes between HPE and no-HPE conditions. This would better illustrate and highlight possible synergies and their driving factors. Lines 259-265: the

analysis looks very tentative and authors give the impression that they speculate. The effect of aerosols on UHI depends on many factors affecting OLR and DLR and deserves some discussion (see for instance Li et al 2020, doi:10.1029/2020EA001100, or Wu et al 2017 (SCIeNTIfIC REPOrTS | 7: 11422 | DOI:10.1038/s41598-017-11705-z), also https://doi.org/10.1016/j.scitotenv.2018.04.254. What about nighttime UHI intensity, which is traditionally higher than daytime UHI but also intensified by absorbed radiation by aerosols.

Line 46 in Supplement: it seems that the cities are shown in Fig S1b, but only the locations of 16 meteorological stations stations are presented in this figure. Also, the 11 air quality stations should be indicated on the map. Section 2.2 in supplement: It is mentioned that temperature observations were derived from one single station (HK) for the identification of HPE, due to minor changes between the stations. It sounds somewhat arbitrary. What 'minor' means. If observed temperature differences are marginal, does it mean that there is no UHI phenomenon in the study area? Some of the 16 meteorological stations are rural as it comes out from Figs S1b and S2. Line 75: relative humidity and wind speed were also used for HPE identification? It is reported that only PM10 data were used due to lack of available PM2.5 data during the study period, but model performance (Table S2b) includes PM2.5 observations during HPEs. Do they refer to another period? The authors use different metrics to evaluate model performance against observations. They provide formulas for three of them, namely r, MB and RMSE. Once they decided to provide formulas for these 3 metrics, they should provide formulas for all metrics, including index of agreement. Also, please check the equations in Supplement for MB, RMSE. The formula for MB is apparently wrong (I do not understand what the symbol '+' represents here). The same is true for the formula 3 for RMSE (what ...j=1, M... represents here?). Despite the wrong formulas included in the text, I want to believe that calculations and presented results based on these metrics are correct, but the authors should check it! Table S2b: it is RMSE and not RSME! Table S2b: why MB for Relative humidity is negative since Model Mean is higher than Obs Mean? Supplement, line 107 & Table S3. : 'The air quality

was considered adequate because the PM10 and O3 thresholds exceed the WHO acceptable annual and daily (Anon, 2006).'. This is confusing. First, rephrase, and then please provide more quantitative information about exceedance thresholds (annual or daily). What 'annual' means here since you used only summer-autumn data. What the 'mean' values (percentiles) in Table S3 represent? Mean daily values for the 11 regional stations? Only for HK stations? Please specify and add corresponding WHO thresholds . Line 136 (supplement' : what do you mean by 'did not vary significantly?' . Please specify/rephrase Technical corrections Figures 1, 2: change/improve color palette in line plots, it is difficult to discern variables Line 20 : replace 'was' with 'were' Line 23: please rephrase Line 34: replace 'more seriously' Line 38: replace 'heat wave' with 'heat waves' Lines 43, 46: replace 'S. Fan' with 'Fan' Line 46: 'these events'

---

## Referee Comment (RC2) · Anonymous Referee #2 · 2 Jan 2021

General comments: This study used observational data and numerical model simulations to identify eight HPEs during 2009–2011, and group the HPEs into three clusters (TC-HPE, ST-HPE, HY-HPE) based on their characteristics and mechanism. The relationship between HPEs and UHIs, air pollution during these episodes was also examined. This study has some interesting findings. My recommendation is to accept with major revision. Some concerns that need to be addressed: 1. Figure S2: I suggest authors to add the location of both all meteorology stations and PM stations into

figure S2, which helps to understand the underlying surface information of the areas where stations located. 2. Table S4: Table S4 is important information for this study, I suggest to move this table from supplement to manuscript. 3. Fig. 1 and Fig. 2 are repeated, please check and revise. And authors should specify the results from observations or model simulations for each figure captions. 4. Line 161: "...cloud fraction (CF; yellow)...", the line color of CF is green instead of yellow. 5. Line 168: Please revise "HPC_TC" to "TC-HPC". 6. Lines 239-240 show "...(d-f) Vertical profile of potential temperature ($\theta$) for TC-HPErep, ST-HPErep, and HY-HPErep at mid-day for the PRD region...", but lines 244-245 show "...Figure 2d depicts the difference between the potential temperature over urban and vegetated land covers in the PRD region at mid-day...", Please clarify whether Fig. 2(d-f) are "potential temperature..." or "the difference between the potential temperature over urban and vegetated land covers...". 7. Discussion: Do authors have any suggestions or thoughts about improving air quality (such as O3-PMx synergistic governance) in summer in the PRD region based on your research. If so, welcome to add them into discussion.

---

## Author Response (AR1)

Dear Editor and Reviewers,

  Thank you very much for your effort on this review and valuable comments and suggestions. We have addressed all the comments one by one and incorporated suggestions in the revised manuscript. Hope you find our revisions useful. Thank you again.

--
Regards,

Steve
* * *
YIM, Hung-Lam Steve, Ph.D.

Associate Professor
Department of Geography and Resource Management

The Chinese University of Hong Kong, Shatin, Hong Kong
Tel: (852) 3943 6534
Fax: (852) 2603 5006
Email: steveyim@cuhk.edu.hk
GRMD@CUHK: http://www.grm.cuhk.edu.hk/eng/

**Anonymous Referee #2**

**General comments: This study used observational data and numerical model simulations to identify eight HPEs during 2009–2011, and group the HPEs into three clusters (TC-HPE, ST-HPE, HY-HPE) based on their characteristics and mechanism. The relationship between HPEs and UHIs, air pollution during these episodes was also examined. This study has some interesting findings. My recommendation is to accept with major revision. Some concerns that need to be addressed:**

**1. Figure S2: I suggest authors to add the location of both all meteorology stations and PM stations into figure S2, which helps to understand the underlying surface information of the areas where stations located.**
Response: Thanks for your suggestion. Figure s2 has a lot of information on it already. Adding station information on it may make it more difficult to be understood. We therefore added the locations of the air quality stations to figure s1c, as we believe

**2. Table S4: Table S4 is important information for this study, I suggest to move this table from supplement to manuscript.**
Response: Your suggestion has been noted and table S4 is moved to the manuscript.

**3. Fig. 1 and Fig. 2 are repeated, please check and revise. And authors should specify the results from observations or model simulations for each figure captions.**
Response: Thanks for the comment, the captions of the figures have been updated as per your suggestion.

**4. Line 161: ". . .cloud fraction (CF; yellow). . .", the line color of CF is green instead of yellow.**
Response: Thanks for your observation, the correction "cloud fraction (CF; cyan)" has been made as suggested.

**5. Line 168: Please revise "HPC_TC" to "TC-HPC".**
Response: Thanks for the correction, "HPE_TC" is changed to "TC-HPE"

**6. Lines 239-240 show ". . .(d-f) Vertical profile of potential temperature ($\theta$) for TC-HPErep, ST-HPErep, and HY-HPErep at mid-day for the PRD region. . .", but lines 244-245 show ". . .Figure 2d depicts the difference between the potential temperature over urban and vegetated land covers in the PRD region at mid-day. . .", Please clarify whether Fig. 2(d-f) are "potential temperature. . ." or "the difference between the potential temperature over urban and vegetated land covers. . .".**

Response: Thanks for that critical observation, the caption "(d-f) Vertical profile of potential temperature ($\theta$) for…" has been changed to "(d-f) Vertical profile of potential temperature difference ($\theta_{(diff)}$) for…" because they represent the difference between urban and vegetated land covers.

**7. Discussion: Do authors have any suggestions or thoughts about improving air quality (such as O3-PMx synergistic governance) in summer in the PRD region based on your research. If so, welcome to add them into discussion.**

Response: Thanks for this suggestion, however, we did not include it in this study as we focused on understanding the mechanisms responsible for the HPEs but will consider this in our future studies.

Anonymous Referee #3

General comments: The study focuses on the region of Pearl River Region (China) and investigates hot and polluted episodes (HPEs) from 2009 to 2011, classified in relevance to atmospheric circulation, while it explores variability of surface energy balance parameters in an effort to detect possible synergies between HPEs and urban heat island (UHI). Numerical simulations using WRF-Chem were conducted in the domain, after evaluating model performance through observations. The subject of the study is relevant to the thematic areas of the journal. Synergistic effects between heat waves, urban heat islands and air pollution are of particular importance in the general context of climate change and environmental threats. Although the subject of the study is interesting, the aim, specific objectives and novelty of this study should be further highlighted. Interpretation and discussion of the results remains somewhat superficial, especially regarding the role of different surface fluxes during HPE (compared to pre- and post – HPEs episodes) and how they contribute to positive feedbacks between HPEs and UHI strength in the study area. Methodological approaches need further clarification and precision in several parts of the study (in which authors seem to speculate) as for instance adopted criteria for the identification of HPEs and utilization of meteorological and air quality observations from the available stations network in the study area. Limitations of this study should be included in the discussion. Specific comments are provided below, which I hope will be helpful to the authors.

Response: Thank you very much for your valuable comments and suggestions. We addressed them one by one below. Hope you find our revisions useful. Thank you again.

**Specific Comments:**
1.Why the period of 2009-2011 was used? Recent decade has witnessed extreme hot weather and could be more representative of recent and near future tendencies. Also, I missed some qualitative and quantitative information about the observed air quality and temperature trends in the area during the last years. For instance, are heat waves, polluted episodes, or combination (HPEs) becoming more frequent?

Response: Thanks for the comments, the period 2009-2011 was used because of the availability of data needed for the study. We didn't consider the trends of the air quality or meteorology at this time, as the focus of this research is to understand the mechanisms responsible for the occurrence of HPEs. Nevertheless, our further studies will attempt to understand the trend. Thank you for your suggestion.

2. Line 49-50 : actually the outcome of these studies is that asymmetric changes in the surface energy budget equation terms during heat waves (compared to normal summer conditions),, have been found to induce synergies between heat waves and UHI magnitude Line 70 – 75: The title of the manuscript refers to synergies between HPEs and UHIs. But here, the main aim / objectives and intentions of the study are not clearly stated. Line 86-87: what previous study? Please specify and rephrase. If you want to stress the innovation of the 'current ' study which does not duplicate a 'previous' one, just underline what has been already done and what is new in this study.

Response: Thanks for your comments. The study focused on HPEs, which is different from heat waves since it is interested in the combined effect of "hot" and "polluted" days. These terms are defined in section 2.2 of the paper, as they form the basis for selecting HPEs. More so, the study aimed to examine this relationship for the different classes of HPE during our study period. Line 71 – 75 has been rewritten as "Representative episodes were examined for a possible synergistic relationship between the urban/vegetated land covers and the HPE within

the PRD region (see SI section 1 and Figure S2 for details in land cover characteristics, and delineation of urban and vegetated surfaces). This is expected to contribute to advancing knowledge regarding the factors responsible for the evolution and sustenance of HPEs as well as the relationship between HPEs and surface characteristic".

The title includes two main parts: (1) characteristics of surface energy balance and atmospheric circulation during hot-and-polluted episodes, whereas (2) their synergistic relationships with urban heat islands. Our results section is consisted of three parts: (1) HPE identification and classification, (2) characteristics of surface energy balance and atmospheric circulation during identified HPEs, (3) the HPE and UHI interactions. So, the current title is considered as appropriate to reflect our work scopes.

**3.There is a confusion with citations and reference list of Yim. Yim 2020, Yim 2020a and Yim 2020b, all 3 appear in the text. In the reference list they appear as 2020a and 2020b, but these two references are also identical. Please clarify and correct.**
Response: Thank you for the observation. The citations are all corrected to Yim(2020). Thanks again.

**4. The criteria for the definition and methods for identification of HPE should be better presented and/or justified. There is a plethora of heat waves (or hot episodes) definitions in literature based either on percentiles of maximum temperature distribution (reflecting local climate conditions), fixed temperature thresholds or combination. A common duration of 3 days is usually adopted. According to the authors, the 50% percentile of T2 exceeds the critical temperature above which risks of health impact were significant (SI, line 106). Is it for Hong Kong station? Which is this threshold? This means that at a frequency of 50% HK experiences temperatures above critical values? The same for air quality thresholds. Provide some quantitative information. Are WHO thresholds exceeded at a frequency of 50% in the area? Is it for HK stations (being also cleaner than other air quality stations, according to authors?). The authors should proceed for their identification over the study period based on a more robust definition of HPEs. In the sensitivity test (Table S3 in SI) I had the impression that the percentile 50% was selected just to secure a 'good' number of HPEs (not too few, not too many?). What is the measure of the 'optimum' number of HPEs?**
Response: Thanks for your questions, the study used a combination of both threshold and critical values, but since it is not focused on heat waves, the thresholds often used for heat waves cannot be directly applied. The critical temperature was selected based on the critical effect on human health Chan et al., (2012). Above which risks of health impact were significant, and 95[th] percentile derived by (Wang et al., 2019b). For maximum temperature, the study used the June – October months (2009 – 2011), which represents the warm periods in the study area. Hence 50[th] percentile do not refer to the 50% of the time of a year.

**5. In general the authors must be and look more confident about their choices and procedures. Lines 92-95. Here it is reported that for HPEs identification, maximum air temperature exceeded threshold at each station ( I suppose the authors refer to 16 meteorological stations within PRD region, as shown in map S1b) and PM10, O3 exceeded thresholds for 50% of all air pollution stations (I suppose from the 11 air quality in total as also reported in supplement in PRD region) . But in Supplement, (section 2.2), HPEs identification seems to be based only on one station's (Hong Kong) temperature, and on 14 air quality stations within Hong Kong. Perhaps I misunderstood something, but I think it is somewhat confusing.**

Response: Thank you for the observation, to clarify, one station was used for the temperature identification, and justification for the choice given in the SI section 3, while 16 stations were used for the model evaluation. 14 air quality stations were used for the HPE identification, but 3 of the stations were excluded during the model evaluation due to insufficient hourly data needed.

**6. In each case, the approach needs further clarification. Line 97-99: I do not understand the point, it is very general statement. Justify better your selection criteria. Are they more strict or less strict compared to previous studies and why? Computational cost is important, but scientific base is more important. Section 2.1 – observations. The number of stations should be mentioned here along with reference to supplement for more details on locations etc. Line 104: write 'in Supplement Information' instead of SI when first mentioned in the text.**

Response: Thank you for the suggestions, they have been taken into consideration and this section have been rewritten as follows:

*The time period was selected because these months represent a period in a year with the highest daily maximum temperature, which when combined with days with poor air quality, forms a HPE. More so, Chan et al., (2012) and Wang et al., (2019b) identified 28.2 °C and 29.77 °C as critical temperature for Hong Kong above which the risk of heat related illnesses increases. $T_2$, $O_3$ and $PM_{10}$ thresholds were first defined as their $50^{th}$ percentiles at each station in the study period. The maximum temperature data at Hong Kong Observatory (https://www.hko.gov.hk/en/cis/stn.htm) – HKO weather station (latitude: 22°18'07"; longitude: 114°10'27") was used for the HPE identification. The justification for the use of the station is presented in Supplementary Information (SI) section 3. A total of 14 air quality stations for the $PM_{10}$ and $O_3$ were used for the HPE identification.*

Further clarification on the difference between Yim (2020) and present study is also rewritten as:

*It is because this model study was designed to combine the methods for the traditional heatwave definition (occurrence for an extended period) and health impact study requiring the use of a critical value.*

This statement also helps justify the HPE identification method used.

**7. Line 106: provide links**
Response: Thanks for your suggestion, the link to the dataset has been added.

**8. Line 115: 'trends' ?**
Response: Thanks for your suggestion, the trends have been changed to "variations"

**9. Line 118: rephrase**
Response: *Thank you for the suggestion, the sentence has been rewritten as "The model results for all the CTRL episodic simulations were separated into groups using K-means clustering, and a representative of each group was characterized based on the mechanisms responsible for its formation"* The two simulations were named and the set of simulations used for this analysis mentioned "CTRL" for clarity.

**10. Line 120: .. 'and synergistic relationship between HPEs and surface cover': please rephrase the part of 'surface cover'**

Response: Thank you for the suggestion, the sentence has been rephrased as *".. and synergistic relationship between HPEs and UHI effect"*.

**11. Line 199: As you refer to short wave radiation (SW), the values 1,700 and 1,500 W/m2 are unrealistic here, based also on the scale of SW. Section 3.2.3 (HY-HPE): according to criteria set for T2 , the identification of HPE requires T2 > 50% percentile, corresponding to 31.3 0C (as shown in table S3), but simulated temperatures are lower, not corresponding to heat wave criteria.**

Response: Thank you for the observations. The errors made in the values for short wave radiation has been corrected and the sentence now reads *"On day 4, the convective motions (−0.1 m/s) contributed to a steady increase in cloud fraction to 0.5. Consequently, the increase in cloud cover cut off insolation (down to approximately 850 W/m² on day 4 and approximately 680 W/m² on day 5, as shown in Figure 1b). However, the cloud could retain most of the heat on day 4 as the OLR displayed in Figure 1e decreased from 210 W/m² on day 3 to 165 W/m² on day 4, while the $T_2$ increased by 0.6 °C. $T_2$ reduced thereafter by 2.3 °C due to the amount of persistent high cloud, which blocked incoming shortwave radiation and cut off the source of heat. This episode persisted until the end of the ST-HPE$_{rep}$ (day 6). "*

For the observations, in section 3.2.3, it should be noted that the temperature values used in the study are the domain averages and not the values of specific locations, hence the reason why they appear lower than the temperature threshold. The titles of the figures have been modified to reflect this as not to cause this misunderstanding.

**12. Line 233: OLR is mentioned here (also in line 290) but I cannot see how it is used in the analysis and results described in this section.**

Response: Thanks for the comments, the sentence indicating the use of OLR as an indicator in line 233 has been removed.

**13. Line 234: How were UHI intensities quantified? Do they correspond to T2 differences between 'average urban' grids and 'average rural' grids? (the same for sensible and latent heat differences). How do they link with map of Figure S2? Please provide relevant information. What about observed UHI intensities between urban and rural stations during HPEs? Provide an example. Are there synergies?**

Response: Thank you for the questions. To clarify, the following has been added to section 3.3 "The UHI intensities were quantified as the difference between the average urban and rural grids (as shown in Figure S2). The rural grids are the average of all the vegetated land use land cover categorizations shown in Figure S2. Quantification of potential temperature, sensible and latent heat fluxes were also carried out using the same method."

Unfortunately, we did not study urban and rural differences at the stations level, but rather contrasted the contributions of the different land uses to the HPE for the entire domain.

**14. Line 238 (figure caption): replace 'Difference of T2(diff)' with 'Difference of T2 (T2(diff))'**

Responses: Thank you for the suggestion, they have been modified as suggested.

**15. In the analysis and investigation of possible 'synergies' between UHIs and HPEs, you should better and further emphasize the 'differences' of UHI magnitude between HPE and no-HPE conditions (before and after the episodes). If and how UHI intensity is exacerbated (or possibly attenuated) during the episodes and how it links with possible**

corresponding differences in shortwave radiation, latent or sensible heat fluxes between HPE and no-HPE conditions. This would better illustrate and highlight possible synergies and their driving factors.

Response: Thank you for the suggestions, the following statements have been added to line 237, and it now reads

"The results indicate a remarkable 4.2 °C to 5.5 °C difference in $T_2$ between the two types of land cover due to the UHI effect during the HPE, while the pre- and post- HPE UHI effect had a lower range of 2.2 °C to 4.2 °C, indicating the contribution of the HPE to the UHI effect ." as per your suggestions.

Lines 249 -253 have also been rewritten as

"Although both the latent and sensible heat fluxes shared similar incoming shortwave radiation, the continued increase in the temperature can be attributed to the continued desiccation of the urban areas, leading to faster buildup of heat and increase in temperature, even as the cloud cover continued to decrease until day 6, marking the end of the episode." as per your suggestions.

The following statements were also added to section 3.2.3

"This contrasts with the pre/post periods which had a minimal difference in their cloud fraction during the episode, hence highlighting the importance of incoming solar radiation to the HPE. The sensible and latent heat fluxes shown Figure 2b indicate that the heat buildup for this episode started before the onset of this HPE, as the continuously increased (decreased) for sensible (latent) heat. However, the changes in the cloud fraction from day 4 and 5, led to the retention of the accumulated heat, and subsequent attenuation of temperature, sensible and latent heat differences."

**16. Lines 259-265: the analysis looks very tentative and authors give the impression that they speculate. The effect of aerosols on UHI depends on many factors affecting OLR and DLR and deserves some discussion (see for instance Li et al 2020, doi:10.1029/2020EA001100, or Wu et al 2017 (SCIeNTIfIC REPOrTS | 7: 11422 | DOI:10.1038/s41598-017-11705-z), also https://doi.org/10.1016/j.scitotenv.2018.04.254. What about nighttime UHI intensity, which is traditionally higher than daytime UHI but also intensified by absorbed radiation by aerosols.**

Response: Thank you so much for the citations and your suggestions, they were very insightful. Nevertheless, we did not seek to examine the impact of aerosols on the UHI intensity at this time as it is out of our study scope. Our study was more interested in the overall effect of total aerosol radiative forcing for the entire domain during the HPEs. We will attempt to address your suggested topic in detail in our future study.

**17. Line 46 in Supplement: it seems that the cities are shown in Fig S1b, but only the locations of 16 meteorological stations stations are presented in this figure. Also, the 11 air quality stations should be indicated on the map.**

Response: Thank you for your suggestions, Figure S1, has been to modified as suggested. Figure S1a, shows the WRF-Chem downscaling map; S1b shows the location of all the cities while S1c shows the locations of the meteorology and air quality stations.

**18. Section 2.2 in supplement: It is mentioned that temperature observations were derived from one single station (HK) for the identification of HPE, due to minor changes between the stations. It sounds somewhat arbitrary. What 'minor' means. If observed temperature differences are marginal, does it mean that there is no UHI phenomenon in**

**the study area? Some of the 16 meteorological stations are rural as it comes out from Figs S1b and S2.**

Response: Thank you for your questions. Using the HKO station was because of three reasons. First, as stated in line 96, the temperature difference among urban stations was small. Second, the HKO station is located in the downtown of HK and thus serves as a representative urban station in Hong Kong. Third, the critical temperature was chosen for the HPE identification because medical studies have reported the risk of health impact above the critical temperature. While population is concentrated in urban areas, using the HKO station was considered as adequate in this study.

**19.Line 75: relative humidity and wind speed were also used for HPE identification? It is reported that only PM10 data were used due to lack of available PM2.5 data during the study period, but model performance (Table S2b) includes PM2.5 observations during HPEs. Do they refer to another period?**

Response: Thank you for the question. Relative humidity and wind speed were not used for the HPE identification, rather for the model evaluation. Some stations did not have sufficient PM 2.5 data to make it efficient variable that can be used for the HPE identification.

**20. The authors use different metrics to evaluate model performance against observations. They provide formulas for three of them, namely r, MB and RMSE. Once they decided to provide formulas for these 3 metrics, they should provide formulas for all metrics, including index of agreement. Also, please check the equations in Supplement for MB, RMSE. The formula for MB is apparently wrong (I do not understand what the symbol '+' represents here). The same is true for the formula 3 for RMSE (what . . .j=1, M. . . represents here?). Despite the wrong formulas included in the text, I want to believe that calculations and presented results based on these metrics are correct, but the authors should check it! Table S2b: it is RMSE and not RSME! Table S2b: why MB for Relative humidity is negative since Model Mean is higher than Obs Mean?**

Response: Thank you for the comments. The formulae have been corrected as per your suggestion. The results have been rechecked and they are accurate. The metrics such as NMB, MNB and MFB that was not used in the discussion have also been removed, this will also ensure more uniformity between the two tests. The negative value is because the model results were lower than the observations.

**21. Supplement, line 107 & Table S3. : 'The air quality was considered adequate because the PM10 and O3 thresholds exceed the WHO acceptable annual and daily (Anon, 2006).'. This is confusing. First, rephrase, and then please provide more quantitative information about exceedance thresholds (annual or daily). What 'annual' means here since you used only summer-autumn data. What the 'mean' values (percentiles) in Table S3 represent? Mean daily values for the 11 regional stations? Only for HK stations? Please specify and add corresponding WHO thresholds.**

Response: Thanks for your comments. The following changes will made as per your comments; line 107 was changed to read "quality was considered adequate because the $PM_{10}$ threshold exceed the WHO acceptable annual mean (20 $\mu g/m^3$) (Anon, 2006)". Lines 111 – 113 and the table S3 were also replaced with "50th percentile which represents the median value was also adopted as it represents the middle value in the distribution without the interference of large outliers in the distribution that could be a problem if the mean value was to be used. This ensured that a true statistical midpoint in the distribution of the variables was used as the threshold for the identification of the HPEs". Only Hong Kong stations were used for the

identification as they served as they were the only data available for this study during that period.

**22. Line 136 (supplement' : what do you mean by 'did not vary significantly?'.**
Response: Thank you for the question. The statement has been modified to read "Particularly, $T_2$ among the HPEs in each group did not vary significantly, whereas remarkable differences were found in $T_2$ among the three groups."

**23. Please specify/rephrase Technical corrections Figures 1, 2: change/improve color palette in line plots, it is difficult to discern variables.**
Response: Thank you for your suggestion, the color palette has been modified as suggested.

**24. Line 20 : replace 'was' with 'were'. Line 23: please rephrase Line 34: replace 'more seriously' Line 38: replace 'heat wave' with 'heat waves' Lines 43, 46: replace 'S. Fan' with 'Fan' Line 46: 'these events.**
Response: Thank you for pointing out these technical issues, the changes have been made as per your suggestion.

---

## Author Response (AR2)

**The authors prepared a new version of the manuscript including corrections and clarifications on a number of issues, as suggested.**

**1. Despite their efforts my main concern regarding the robustness and clarity of HPE identification remains. The criteria and methodology for HPE identification are described and presented in a somewhat messy and patchy way in the manuscript. Some information is included in the main text. Another short paragraph is included in Supplement, but without much information. It was difficult to reach and combine all this information. The thresholds for O3 are not included, although O3 has been included in the criteria for HPEs. The annual threshold value of 20µg/m3 is used for PM10, although the authors assume daily values (50th percentiles) for at least 3 consecutive days and the daily threshold of WHO is 50 µg/m3. I suggest the authors to gather and include all information and criteria for HPE identification in a section and describe them in a precise way giving also quantitative information when needed. For instance**

**An HPE was identified when the following criteria were simultaneously fulfilled:**

**i. Daily Tmax at HK observatory was above 50th percentile of Tmax distribution from June-October over the study period 2009-2011 for at least 3 consecutive days. (which is this value? And comment that it is at the same time above the health risk values found by Chen et al, or Wang et al**

**ii. 50th of daily PM10 concentration was above (provide threshold) for 3 consecutive days at 50% of the 11 air quality stations at Hong Kong (?)**

**iii. 50th of daily O3 was above (xxxx…provide a threshold) for 3 consecutive days at 50% of the 11 air quality stations at Hong Kong (?)**

Response: Thank you for your suggestions, they are highly appreciated. Section 3 of the supplementary material have been removed and merged with the section 2.2 in the manuscript, and it has been rewritten to reflect your suggestions.

"The concept of hot polluted episodes that refer to an episode with coincident high temperature and air pollution level have been investigated (Chan et al., 2012; Katsouyanni et al., 1993; Ordóñez et al., 2010; Yim, 2020), however, most of the studies have been focused on their compounding impact on health (Chan et al., 2012)  or the impact of the excess temperature on air pollution levels (Ordóñez et al.,

2010). Only few studies (Fan et al., 2011; Yim, 2020) have considered the mechanisms responsible for their co-occurrence. While Yim (2020) focused on high temperature and PM2.5 pollution in Hong Kong, this study focused on high temperature and high PM10 and O3 in the PRD region. Similar to, Yim (2020), study identified a HPE based on daily maximum temperature, and daily mean of PM10 and O3. PM10 was used in this study because of the lack of PM2.5 data in the study period which was June to October in 2009–2011. The study period was selected because these months represent a period in a year with the highest daily maximum temperature, which when combined with days with poor air quality, forms an HPE.

The HPE identification took into account the methods for the traditional heatwave definition (occurrence for an extended period and threshold), and health impact study requiring the use of a critical value (Chan et al., 2012; Wang et al., 2019b). Thresholds for temperature, PM10 and O3 were set as the mean of 50th percentiles of the variables of all stations, i.e. 31.3 °C, 31 μg/m3 and 24 μg/m3, respectively. The thresholds represented the middle value of their distributions without the interference of outliers in the distribution. Hence, a HPE was identified if the following conditions are fulfilled simultaneously:

The daily maximum temperature of Hong Kong Observatory (HKO) station (22°18'0''N, 114°10'2''E) exceeds the temperature threshold for 3 consecutive days, and

Daily means of PM10 and O3 exceed their thresholds for 3 consecutive days.

Daily maximum air temperature data was obtained from the HKO station was used for the HPE identification with three reasons. First, the temperature difference among urban stations was marginal such that one urban station should be fine to represent the overall temperature of urban areas in Hong Kong. Second, the HKO station is in the downtown of Hong Kong and thus serves as a representative urban station in Hong Kong. Third, the critical temperature was chosen for the HPE identification because epidemiological studies have reported the risk of health impact above a critical temperature (28.4 °C) which is lower than our temperature criteria, meaning that the health impact during our identified HPEs should be expected to be more adverse. While population is concentrated in urban areas, using the HKO station was considered as adequate in this study.

The daily mean data for O3 and PM10 used for HPE identification were obtained from 14 air quality stations within Hong Kong. These were operated by the Hong Kong Environmental Protection Department during the study period. Hong Kong air quality stations were used as proxies to identify HPEs in the PRD region because there was insufficient data for the stations in the region during the study period. The PRD regional air quality monitoring network's annual and quarterly report between 2013–2018 showed that air quality in Hong Kong was always better than or similar to the other stations in the region (Anon, 2019)"

**2. If the criteria for air pollution are not strict please comment on this and provide an explanation for this decision. Also, a short period was selected for this study (2009-2011) due to lack of available data for a longer period and more recent period. But even in this short period, several assumptions were made due to lack of data in some stations. The limitations of the study should be discussed.**

Response: Thank you for your suggestion. A section for the limitations of the study has been added as per your suggestion. A long term study period is beyond the scope of this study because this study was meant to explore and understand the mechanisms responsible for different categories of HPEs, and other studies that are focused on understanding mechanisms and characteristics of HPEs suggest that 3 years is sufficient.

Limitations of the Study: This study was limited by access to more recent air quality emission data; hence the study was conducted for the 2009 - 2011 period. The use of data from only Hong Kong area as a proxy for the region was because of the lack of air quality data from the rest of the study area. Also, as mentioned in section 2.2, PM10 was used in this study because of the inadequacy of PM2.5 data in the study period.

**Additional comments**

**3. Although in their response to my question, the authors clarify that relative humidity and wind speed are used only for model simulations and not for HPE identification, in section 2.2 in**

**Supplement entitled 'Observational Data for HPE Identification' wind speed and relative humidity are also included.**

85 Response: Thank you for your comment, the section has been rewritten as follows:

"Daily maximum air temperature data was obtained from the HKO station was used for the HPE identification with three reasons. First, the temperature difference among urban stations was marginal such that one urban station should be fine to represent the overall temperature of urban areas in Hong Kong. Second, the HKO station is in the downtown of Hong Kong and thus serves as a representative

90 urban station in Hong Kong. Third, the critical temperature was chosen for the HPE identification because epidemiological studies have reported the risk of health impact above a critical temperature (28.4 °C) which is lower than our temperature criteria, meaning that the health impact during our identified HPEs should be expected to be more adverse. While population is concentrated in urban areas, using the HKO station was considered as adequate in this study.

95 The daily mean data for O3 and PM10 used for HPE identification were obtained from 14 air quality stations within Hong Kong. These were operated by the Hong Kong Environmental Protection Department during the study period. Hong Kong air quality stations were used as proxies to identify HPEs in the PRD region because there was insufficient data for the stations in the region during the study period. The PRD regional air quality monitoring network's annual and quarterly report between

100 2013–2018 showed that air quality in Hong Kong was always better than or similar to the other stations in the region (Anon, 2019)."

**4. In their answer of comment No 20, the authors insist that the model results of RH are lower than the observations and thus Mean Bias is negative. But Table S2a shows the opposite. If indeed model value is lower than the observed value, then Table S2a must be corrected.**

105 Response: Thank you for your observation, the correction has been made to Table S2a.

**5. Line 247: …' because they exhibit the characteristics of an air parcel over time'.. I do not understand, this is really strange. Please explain or provide a complete sentence that makes sense regarding the suitability of potential temperature as indicator in this case.**

Response: Thank you for pointing this out, to clarify, the sentence was rewritten as follows:

[revised manuscript text omitted]